# Effects of Different Feeding Regimes on Growth Rates and Fatty Acid Composition of Largemouth Bass *Micropterus nigricans* at High Water Temperatures

**DOI:** 10.3390/ani12202797

**Published:** 2022-10-17

**Authors:** James T. Tuttle, Matthew A. Smith, Luke A. Roy, Michele Jones, Rebecca Lochmann, Anita M. Kelly

**Affiliations:** 1School of Fisheries, Aquaculture & Aquatic Sciences, Alabama Fish Farming Center, 529 South Centreville Street, Greensboro, AL 36744, USA; 2Department of Aquaculture and Fisheries, University of Arkansas at Pine Bluff, 1200 North University Drive, Pine Bluff, AR 71601, USA; 3College of Food, Agricultural, and Environmental Sciences, Ohio State University, South Centers, 1864 Shyville Road, Piketon, OH 45661, USA

**Keywords:** fingerling, feed conversion ratio, thermal stress, satiation, lipid extraction, feed regime, fatty acid profile

## Abstract

**Simple Summary:**

The Largemouth bass aquaculture industry is growing in the United States. Many producers are shifting toward raising small juvenile fish, referred to as fingerlings. During the summer months, when water temperatures are highest, producers need to know how to best feed their fingerlings to ensure the best growth. This study aimed to determine which feeding strategies and feed amounts can result in healthy fish growth with a desirable fat content while under heat stress. The results of this study indicate that feeding fingerlings a lower amount daily or feeding every other day will result in the most efficient growth. These results also allow producers to form an economic feeding plan for fingerlings when water temperatures are high.

**Abstract:**

As the northern Largemouth bass (LMB) (*Micropterus nigricans*) industry shifts toward fingerling production, implementing practical feeding strategies to ensure efficient growth during high water temperatures is paramount. Twenty (12.7 ± 0.2 g) (Trial 1) and fifteen (7.2 ± 0.1 g) (Trial 2) LMB fingerlings were stocked in two recirculating systems (each containing nine tanks), acclimated to 30 °C, with one system fed daily rations of 3, 5 and 7% body weight (Trial 1), and the second system fed to satiation daily, every second day, or every third day (Trial 2), for 28 days each. All treatments were triplicated. Multiple growth metrics and lipid composition were analyzed. The 3% treatment yielded the lowest final average weight (36.05 g) and FCR (0.83), with no difference in final biomass in Trial 1 treatments. Fish fed to satiation daily and every second day produced FCRs and biomasses of 0.83 and 356.78 g, and 0.93 and 272.26 g, respectively. There were no differences in total lipid concentration, however, fatty acid profiles differed significantly between all treatments within their respective trials. Feeding LMB fingerlings 3% of total body weight or feeding daily to satiation allows for efficient growth at 30 °C and implements cost-effective feeding strategies.

## 1. Introduction

Due to their ferocious demeanor and large adult size, northern Largemouth bass (*Micropterus nigricans* [1], formerly *Micropterus salmoides salmoides*; hereafter referred to as LMB) are among the most desirable sportfish in the United States (USA). State and federal hatcheries have reported culturing LMB for over a century as part of the stock enhancement programs for natural water bodies [2,3,4,5,6]. Increased feed costs, electricity, labor and the potential for catastrophic disaster are some of the many reasons stocking fingerlings is considerably cheaper than raising larger fish on farms. State and federal stocking enhancement programs also prefer to stock fingerlings that are easier to handle and transport. Therefore, the early life stages of LMB culture have historically been the primary focus of many state and federal hatcheries.

However, LMB research has substantially increased as food-sized (>500 g) fish continue to gain in popularity [7]. Multiple studies have focused on extensive LMB nutrition and alternatives in formulated diets since the late 1990s, including macronutrient requirements [8,9,10], amino acid and fatty acid nutrition [11,12,13,14], and a study on the elimination of marine products from [15] formulated commercial feed. Certain long-chain polyunsaturated fatty acids (LC-PUFA) reportedly provide immunological and nutritional support for LMB, as well as numerous human health benefits, which has led to more in-depth fatty acid analysis in current aquaculture nutrition studies [14]. In addition, pellet characteristics (floating vs. sinking and 5.5 mm size vs. 13.0 mm size) have been investigated to determine their effect on growth and feed conversion efficiency [16]. Furthermore, a vast majority of all food fish LMB raised in the USA are delivered live to ethnic markets in highly populated north American cities [17]. Finally, the feasibility of a filet market [18] has also been investigated, although it is not currently economical.

Other recently published LMB work has shifted focus to indoor recirculating aquaculture systems (RASs) to improve production and shorten growing seasons. Some studies reported a deliberate feed transition regime from LMB fry to fingerlings can result in survival rates up to 70% [19,20,21] and investigated the possibility of shortening the grow-out time by rearing small (average 9.0 g) and advanced (average 36.7 g) fingerlings indoors at various densities. They found that small LMB juveniles can be raised to 30 g in weight with a gross yield of 100–125 kg/m^3^, and advanced LMB could be raised to a maximum yield >70 kg/m^3^. Additionally, Watts et al. [22] found no significant survival, average harvest weight, or feed conversion ratio (FCR) differences as a result of three stocking densities (30, 60, or 120 fish/m^3^) of juvenile LMB after six months. These studies have shown the value of decreasing the culture period by utilizing indoor facilities since the first growing season is relatively short in many parts of the US. First-year post-feed habituation LMB fingerlings raised in commercial ponds may experience water temperatures of nearly 40 °C during the summer months, which is the warmest water temperature to date across all LMB studies [23]. Additionally, an extended culture period (2–3 years) means adult LMB are subject to these elevated temperatures for multiple summers before reaching market size.

Personal communication with producers has revealed that in recent years there are often 4–8 weeks during the hottest months of summer when individual growth stalls or decreases due to the stress at high pond water temperatures. Consequently, this has prompted interest in utilizing alternative management strategies compared to daily satiation feeding. Although not directly measured in these trials, it is widely understood that fish metabolic rate increases as water temperatures increase [24], even if given ample time to acclimate [25]. Water temperature is the most influential factor affecting growth [26,27]. The total amount of food a fish can consume increases as temperature increases up to a certain point. However, so does the necessary maintenance ration [28]. Growth rate and feed consumption sharply drop once fish become stressed due to elevated water temperatures [29,30]. Critical and chronic thermal maxima of LMB stocks and their hybrids have been recorded as low as 29.2 ± 1.4 °C and high as 41.9 ± 0.51 °C [30]. Commercial fish producers cannot economically manipulate water temperature in a pond setting for an extended period. Therefore, they must work towards improving water quality to limit stress through proper management techniques. The ability of farmers to make a more informed decision on feeding during these summer months should also decrease costs by reducing the amount of wasted feed.

Tidwell et al. [12] investigated the effect of three water temperatures (20, 26, and 32 °C) on growth, survival, and biochemical composition of juvenile LMB. In their study, LMB were fed to satiation twice daily. They found no significant differences in growth and feed conversion efficiency between 26 and 32 °C, although LMB raised at 26 °C were more capable of utilizing feed and dietary protein. Their results determined the optimal growing temperature for LMB fingerlings is 26 °C [12]. Fantini et al. [23] demonstrated that LMB reared at 35 °C were stressed to the point that feed intake and growth were significantly suppressed.

Our study aimed to investigate various feeding regimes at a water temperature of 30 °C which is common during the summer months in Arkansas. The null hypothesis was that there would be no differences in performance of LMB under different fedding regimes at 30 °C. An additional trial was run to investigate the effect of extreme temperatures on second-year LMB. Similar to Tidwell et al. [12], we analyzed the fatty acid composition pre- and post-trial to look for significant differences.

Specifically, the main objectives of the indoor recirculating tank trials were to (1) investigate the effect of three different feeding rates on LMB fingerlings (3, 5, 7%) at 30 °C; (2) investigate the effect of satiation feeding daily, every second day, and every third day at 30 °C.

## 2. Materials and Methods

All animal experiments were approved by the Institutional Animal Care and Use Committee (UAPB2015-02) and conducted according to the guidelines of the United States Department of Agriculture (USDA) Animal Welfare Act Regulations. All trials were conducted at the University of Arkansas at Pine Bluff’s Fish Health Services Laboratory in Lonoke, AR, USA. All systems were bleached and scrubbed clean one to two weeks before stocking. Lights were on a timer and 12 hL: 12 hD throughout both trials.

### 2.1. Trial 1 Experimental System

This 28-d trial was conducted in a RAS with nine square plastic tanks (227 L each; 2600 L total system) in the summer of 2015. Each tank had a center standpipe for overflow and the overflow led to a 608 L sump (0.74 mW × 1.6 mL × 0.5 mH) filled with floating biological media. A centrifugal pump (0.39 kW) then pulled water into an AquaDyne 200 series filter (Hartwell, GA, USA) before passing through an ultraviolet (UV) sterilizer (50 W) and returning to the tanks. Oxygen was supplied to each tank and the sump via a regenerative blower. System backwashes were performed as needed and the system was refilled with dechlorinated city water. All tanks and the sump were covered with insulating foam to reduce system heat and water vapor loss. A 100% water exchange occurred approximately every 42 min in each tank. Submersible heaters (TH-800 model; Finnex, Chicago, IL, USA) were used for acclimating the fish over a seven day period, and to maintain water temperatures at 30 °C.

### 2.2. Trial 1 Experimental Fish and Culture Management

Feed-habituated LMB fingerlings were acquired from a commercial farm (J. M. Malone & Son, Inc., Lonoke, AR, USA) and housed at the laboratory until the start of the trial. Tanks were stocked with 20 LMB (1.24 kg/m^3^); mean (±SD) initial body weight, total length, and fish condition were 12.7 ± 0.2 g, 10.5 ± 0.6 cm, and 0.96 ± 0.05, respectively. Fulton’s condition factor (K) was calculated as K = (W/L^3^) × 100,000, where W is the weight of the fish in g and L is the total length in mm. Once the tanks were stocked, the water temperature was raised from ambient (~27 °C) to 30 °C over 9 d.

A comm’rcially available 3 mm extruded, floating feed, consisting of a 48%:18% protein to lipid ratio (Skretting, Tooele, UT, USA), was provided to the fish four times daily at approximately 0830, 1130, 1330, and 1500 h. Treatment groups were assigned at random (three treatments with three replicates) and rations fed were 3, 5, or 7% body weight per day. These percentages were chosen based on daily feed ration standards from commercial LMB producers. If a large amount of feed was not consumed during an earlier feeding, fish were not fed on the following feedings. Any remaining feed was weighed and subtracted to determine the feed amounts provided during the study. Fish were be sampled weekly to determine total biomass per week, and accurate feed amounts for each respective treatment. This was accomplished by randomly selecting one tank per treatment, without resampling the same tank again until all tanks within the same treatment were sampled, thus reducing stress on fingerlings. The biomass from the sampled tank was used to set the feed amount of the other 2 tanks in Trial 1, within each respective treatment.

### 2.3. Water Quality Measurements

Temperature and dissolved oxygen (DO) were measured daily with an oxygen meter (YSI 55; Yellow Springs Instruments, Yellow Springs, OH, USA). Total ammonia nitrogen (TAN), ammonia nitrite (NO_2_-N), and pH were measured weekly. The pH was measured with a pH pen (pH Tester 30 series, Eutech Instruments, Vernon Hills, IL, USA). TAN and NO_2_-N concentrations were measured using a HACH DR 4000 spectrophotometer using the salicylate and diazotization methods, respectively (HACH, Loveland, CO, USA). Total alkalinity was determined at the start and end of the trial with a HACH test kit (FF-2 model, HACH, Loveland, CO, USA).

### 2.4. Lipid Determination

Ten fish were frozen before the trial started to be used as a reference for lipid and fatty acid contents. At the completion of the 28-d trial, ten fish from each tank (90 fish total) were frozen post-trial; however, only six fish from each replicate were used in lipid and fatty acid analysis. The fish were homogenized, freeze-dried, and stored at −70 °C until analysis. To quantify total lipid and fatty acid profiles, a procedure described by Folch et al. [31], and described in detail in Fantini et al. [23], was used to analyze total lipids from the initial diets and whole-body fish samples.

### 2.5. Trial 2 Experimental System

The same experimental RAS was used in Trial 2 as in Trial 1. The 1000 L RAS consisted of nine rectangular glass tanks (42 L each) and a sump (608 L; 0.74 mW × 1.6 mL × 0.5 mH) filled with biological filter balls. Both the experimental tanks and the sump were fitted with overflow pipes to maintain proper water volumes. A centrifugal pump (0.15 kW) was connected to the outflow of the sump and directed water back to the tanks. The regenerative blower, filter, insulation material, backwashing and water exchange procedures, and water heaters detailed in Trial 1 were all identical to those used in Trial 2.

### 2.6. Trial 2 Experimental Fish and Culture Management

LMB fingerlings were acquired from a commercial farm (Dunn’s Fish Farm, Monroe, AR, USA) in the spring of 2015 and housed on-site until the start of the trial. Juveniles were stocked at 15 fish/47 L tank (0.11 kg/m^3^); mean (±SD) initial body weight, total length, and fish condition were 7.2 ± 0.1 g, 8.2 ± 0.7 cm, and 1.21 ± 0.08, respectively. Once stocked, the temperature was raised and maintained from ambient (~21 °C) to 30 °C over 11 d. Feed brand and nutritional values were the same as Trial 1. Treatments for this trial were assigned randomly in triplicate and designated as tanks fed to satiation every day, every second day, or every third day. During the respective feeding days for each treatment, fingerling LMB were fed ad libitum four times daily at approximately 0830, 1130, 1330, and 1500 h. The weight of an individual feed pellet was determined by averaging the weight of 100 dry pellets. The weight of feed consumed by the fish was determined by weighing a container of feed before and after feeding for each tank. Any unconsumed feed was removed, counted, and weight was determined by multiplying the number of pellets by the average weight for dry pellets. The weight of uneaten food was added to the after-feeding weights to determine the total amount of feed consumed. This ensured that the weight of feed consumed was accurate and that no un-eaten feed interfered with the growth of the fish or negatively influenced water quality in the RAS. In both Trials 1 and *2*, the equipment and protocols used to measure water quality parameters and extract and analyze lipids and fatty acids were the same.

### 2.7. Statistical Analysis

For all experimental trials, statistical analyses were conducted using SAS software 9.4 (SAS Institute, Cary, NC, USA). Any *p* ≤ 0.05 was considered significant in statistical comparisons. For Trial 1 and Trial 2, survival rates, initial weights, final biomass, final average weight, final average total length, weight gain percentage, feed conversion ratio (FCR), and Fulton’s condition factor (K) were analyzed using A general linear model regression analysis, and a Tukey’s honest significant difference (HSD) test to evaluate significant differences in these targeted metrics. In addition, the univariate Shapiro–Wilks statistical test for normality was conducted to ensure all growth data had a normal distribution of residuals. In instances where residual distribution was not normal, subsequent data were arc-sin transformed. FCR was calculated as FCR = (Total Feed Consumed in g/Δ Total Biomass in g). Weight gain percentages were calculated as WG% = (Total final biomass in g/Total initial biomass in g) × 100%.

A one-way ANOVA statistical test was used to determine significant differences in fatty acid profiles of whole-body fish samples. A Step-wise discriminant (SDA) analysis and canonical discriminant analysis (CDA) were used to evaluate whole-body LMB samples fatty acid profiles and optimize the number of fatty acids within specific profile to discriminate among classes, respectively [32,33]. The procedures for SDA and CDA analyses were followed as described by Fantini et al. [23].

## 3. Results

Fish did not show any signs of disease in either Trial 1 or Trial 2.

### 3.1. Trial 1 Growth Metrics

For the entire trial, including the 10-day acclimation period, daily mean (±SD) water temperature and DO remained at 29.77 ± 0.65 and 6.80 ± 0.32 mg L^−1^, respectively. Weekly parameters such as pH, TAN, and NO_2_-N concentration were 7.88 ± 0.29, 0.86 ± 0.26 mg L^−1^, and 0.19 ± 0.099 mg L^−1^, respectively. All statistical values were compiled and given a designation as to their statistical significance from each other. Between the 3, 5, and 7% feed rate treatments, survival rates, initial weight, final biomass, and Fulton’s condition factor (K) values throughout the feeding trial were not significant (Table 1). The final average total length in Trial 1 was not statistically significant (p: 0.0521), but the 3 and 7% treatments were significantly different from each other (Table 1).

Additionally, final average weight, weight gain, and FCR between the 3 and 7% treatments were also significantly different from each other (Table 1). Due to issues with excess food, on the tenth day, feeding times were adjusted from four times daily (0830, 1130, 1330, and 1500 h) to twice daily at 0830 and 1500 h. Individuals fed 3% body weight daily were the only treatment group to produce an FCR below one (0.83), whereas the FCRs in the 5 and 7% treatments were 1.21 and 1.66, respectively.

### 3.2. Trial 1 Lipid Determination

The total lipid concentration of the fish did not differ among treatments, but several differences in fatty acids were observed between treatments (Table 2). Palmitelaidic acid 16:1 was highest in fishes in the 5% treatment (group but did not differ from the 3% treatment). Conversely, the 5% treatment was lowest in Palmitoleic acid, 16:1, n-7. The fish in the 3% and 5% treatments had higher concentrations of 17:1, 18:2, n-6, 18:3, n-3, 20:5, 22:5, and 22:6. These two treatments also had a higher ratio of n-3:n-6 fatty acids. The fish in the 3% and 5% treatments had the most similarities in fatty acids, with the fish from the 7% treatment being an outlier. The 7% treatments had the lowest PUFA and LC-PUFA concentrations and the highest saturated fatty acid concentration. Between two classes, the fatty acid concentration differences in whole-body fish samples from Trial 1 were significant (Wilks’ Lambda < 0.001) (Table 3). The results of the SDA analysis indicated six fatty acids (16:1, 20:3, 20:4, 22:1, 22:5, and 22:6) that were discriminated by class, and thus included in the CDA model (Table 4). Canonical variate one resulted in a cumulative variability of 61.0%, and both variates explained 100% of the model variability (Table 3). Within each variate, class means were then ordered. CAN1 in reference to CAN2, whole body profiles of the 3% treatment fish were maximally separated from the other two treatments along CAN1 (Figure 1). The 5% and 7% treatments were similar, but with the 7% still being separated from the 5%. Once vales were ordered with respect to each variate, it was revealed that CAN1 displayed a strong positive loading for 22:6 fatty acid and a negative loading response for 22:5 fatty acid once all variates were pooled within-standardized canonical coefficients for each variate (Table 4). These differences allowed for clear distinction between the 3, 5 and 7% treatment fish and corresponded to higher concentrations of 22:5 in the 3% treatment and higher 22:6 concentrations within the 3% and 5% treatments (Table 2).

### 3.3. Trial 2 Growth Metrics

For this 29-day trial, the daily water temperature was 30.71 ± 0.79 °C and DO concentration was 6.85 ± 0.29 mg L^−1^. Weekly pH, TAN, and NO_2_^−^-N concentrations were 8.26 ± 0.15, 0.33 ± 0.26 mg L^−1^, and 3.75 ± 4.66 mg L^−1^, respectively. Due to the biological filter cycling, NO_2_^−^-N concentration was 10.25 mg L^−1^ during the first week, then dropped to 0.325 mg L^−1^ by the third week. Centrarchids, including LMB, are not as susceptible to nitrite poisoning issues because of their ability to discriminate nitrite in their blood plasma from other compounds [34]. Brief high nitrite concentrations seemingly did not negatively interfere with the outcome of this study. Between the three treatments, survival rates and initial weights were not significantly different (Table 5).

The growth metrics of final biomass, final average weight, final average length, weight gain, FCR, and Fulton’s condition factor displayed significant differences between the feed satiation treatments (Table 5). Throughout the entire trial, the amount of feed consumed by each treatment was recorded on their appropriate feeding days. On average, the fingerlings fed to satiation daily, every second day, and every third day consumed 5.04%, 6.39%, and 6.41% of their total biomass, respectively.

### 3.4. Trial 2 Lipid Determination

Total whole-body lipid concentration did not differ among the fingerling feed satiation treatments (Table 6). Fish fed every second day had the highest concentration of 16:1, 16:1, n-7, and 20:1, whereas fish fed every third day had the highest concentration of 20:5, 22:5, 22:6, and total long-chain polyunsaturated fatty acids (LC-PUFAs). In Trial 2, differences in whole-body fatty acid profiles were significant (Wilks’ Lambda < 0.001) between two classes (Table 7). Results of the SDA indicated three fatty acids (18:1, n-9, 18:1, n-7, and 22:6) affected class discrimination and thus included in the subsequent CDA model (Table 8). The CDA revealed that 96.0% of the variability was attributed to the first canonical variate, and that 100% of the variance was due to the first two canonical variates (Table 7). Once variate class means were properly ordered, CAN1 with respect to CAN2 was revealed to maximally separated whole-body profiles of the fish fed every second day from the other groups along CAN1 (Figure 2). Fish fed daily and every third day were remarkably similar. The positive loading of 22:6 was the most influential in the separation (Table 8), which corresponds with the differences in the concentration of 22:6 in the whole-body analysis.

## 4. Discussion

Subjecting LMB fingerlings to different feeding strategies, at 30 °C, resulted in noticeable effects on growth as well as fatty acid profiles. This study also provided evidence that certain feed regimes can result in effective LMB growth despite thermal stress. These results are similar to those reported by Tidwell et al. [12]. However, when subjected to higher thermal stress, food consumption and growth in LMB declines [23]. Generally, there is a direct relationship between food offered and the growth of juvenile fish at 30 °C. This determination is both simple and nuanced, as producers have options when choosing the best course of action when LMB experience high temperatures. While we recognize that different feeding amounts would result in different growth rates, we used the feeding rates (3, 5, and 7%) currently used within the industry. The intent of the study was to determine the feeding rate, as a percentage of body weight per day, that resulted in the best growth. However, feeds represent a significant amount of the budgets for aquaculture operations. Therefore, implementing a lower feeding rate can result in ideal growth and should increase the economic viability of production. While producers could feed 7% or more to their fish, they are feeding excess amounts of feed that are not resulting in significant additional growth.

During the feed satiation trial, fish were offered feed until consumption stopped. An experimental FCR of 1.5 was used to calculate the predicted increase in biomass and ultimately predict the amount of feed required to obtain the 5% feeding rate for each tank within each treatment. By the conclusion of the trial, it was apparent that a population of LMB raised at 30 °C will consume 5.04%, 6.39%, and 6.41% of their total biomass if fed to satiation daily, every second day, and every third day, respectively. The feeding strategies used during the satiation trial, could also result in different growth rates. However, increased feed consumption by fish fed every second day and every third day, did not result in increased growth. Daily satiation feeding would also provide evidence of the best percentage to feed LMB daily. The 5% feeding rate found in Trial 1 was similar to the 5.04% in the daily satiation feeding treatment.

Lipids are readily used for energy through fatty acid catabolism, also called β-oxidation [35]. In general, lipids can produce 9.45 kcal of useable energy per gram, have digestibility of 85% or more [36], and are the most effective compound at storing surplus energy [37]. This energy can be used in the liver, heart, red muscle, and white muscle [35,38]. Saturated and monounsaturated fatty acids are readily catabolized, but PUFAs are variable and more complicated [35]. Hybrid red tilapia (*Oreochromis mossambiscus* X *O. niloticus*) had an increase in PUFAs during short-term starvation [39]. The total LC-PUFAs in this study were higher in the fish fed every third day, and the fish fed 3% of their body weight. The short-term starvation period could have led to increased LC-PUFA and n-3 concentration in the whole-body fish. In a study with Sharpsnout seabream (*Diplodus puntazzo*), there was an increase in the n-3:n-6 ratio in fish with reduced feeding; the food restriction seems to trigger the preservation of essential fatty acids [40]. A similar effect was noted in Rainbow trout (*Onchorynchuss mykiss*) when feed rations were reduced. PUFAs increased as rations decreased [41].

With respect to global aquaculture, FCR values across multiple finfish species can vary from as low as 0.7 in salmonids to as high as 2.9 in catfish [42]. Speaking specifically on LMB, there is very little information that documents the growth efficacy of this species in a commercial setting while being fed a commercial diet at this experimental temperature. For example, while evaluating the growth success of juvenile LMB, Huang et al. [43] recorded FCR values between 0.88 and 1.21 across eight different artificial feeds with varying nutrients and protein compositions in a recirculating system averaging a water temperature of 25.6 °C. Another study [44] evaluated feed enhancement components for juvenile LMB at 29.6 °C over eight days but did not record growth or FCR data. Concerning experimental data on commercial feed, one study [45] reported that LMB juveniles fed a commercial diet (Classic Bass^®^, extruded, floating; protein/fat: 48/18; Skretting, Tooele, UT, USA) yielded an FCR of 1.06 in a recirculating system at approximately 28 °C for eight weeks. Another study by Tidwell et al. [46] reported FCR values from 1.9 to 2.4 during the summer months, when temperatures were between 30 and 20 °C from May to August, for LMB juveniles cultured in 120.04 ha ponds on test diets.

The FCR results from both the feed rate trial (Trial 1) and the feed satiation trial (Trial 2) were encouraging as they agree with FCRs reported in past LMB growth and nutrition studies. The results also indicate that LMB can still efficiently increase in size without sacrificing health during periods of higher thermal stress. Feeding LMB fingerlings to satiation daily or every other day result in more efficient and significantly higher growth metrics than feeding every third day. Feeding 3% daily when water temperatures reach 30 °C still allows for the most efficient growth of LMB fingerlings, while feeding 5% and 7% can also result in acceptable growth. This information from this study is also helpful to producers and provides options for developing more economically beneficial feed management plans. On average, formulated feeds are the largest financial variable cost in the aquaculture industry, accounting for up to 70% of an operation’s budget [47]. The current average LMB formulated feed cost is between USD1400 and USD1500 per ton [48]. Results from this study indicate that it is more cost-effective for producers to offer less pelleted feed to LMB over a slightly longer period while also experiencing healthy growth in their stocks when water temperatures are approximately 30 °C.

## 5. Conclusions

This study provided insight into how alternative feeding strategies at higher water temperatures influence growth parameters and lipid composition in LMB juveniles. A reduction in feeding can alter the fatty acid composition with the omega 3 essential fatty acids being reserved. Our findings indicate that feeding 3% body weight daily or feeding to satiation every other day provides viable options for LMB fingerlings in culture settings to experience healthy growth in the water temperature used in this study. Further research is necessary to determine if these findings are transferrable to LMB raised on commercial farms, and if there is a need to adjust the feeding regime further when water temperature exceeds 30 °C. During the growing season, when water temperatures reach 30 °C, it is imperative to have warmer water feed management regimes that lead to success for commercial producers who raise LMB juveniles to market-sized adults successfully.

## Figures and Tables

**Figure 1 animals-12-02797-f001:**
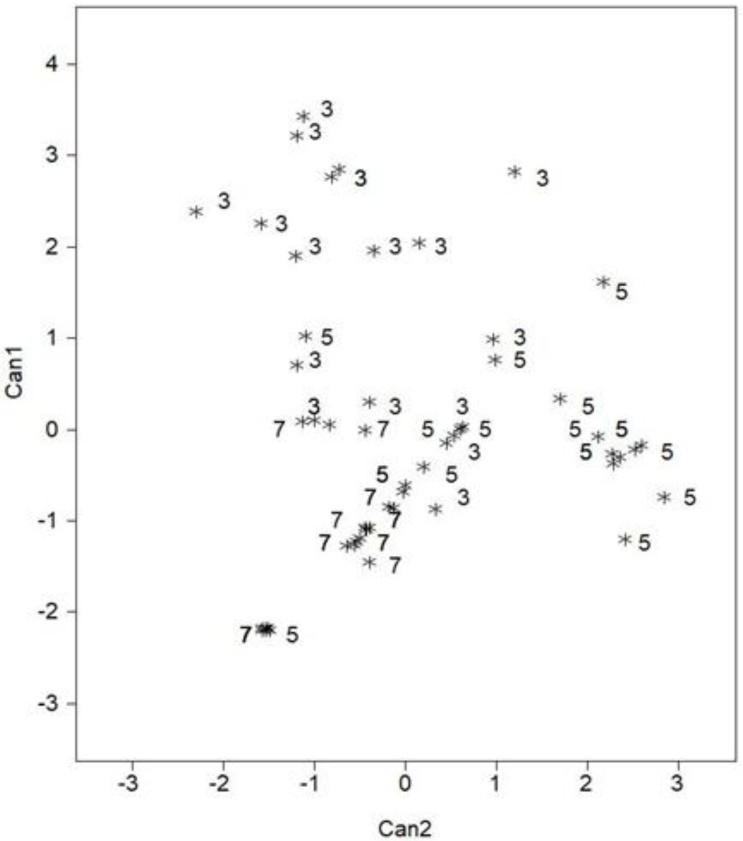
Scatter plot of centroids (class means) for the first and second canonical variates separating the fatty acid profiles of the northern Largemouth bass whole-body samples by diet (Trial 1). Treatments designated as 1: Fed 3% (3), 2: fed 5% (5), and 3: fed 7% (7).

**Figure 2 animals-12-02797-f002:**
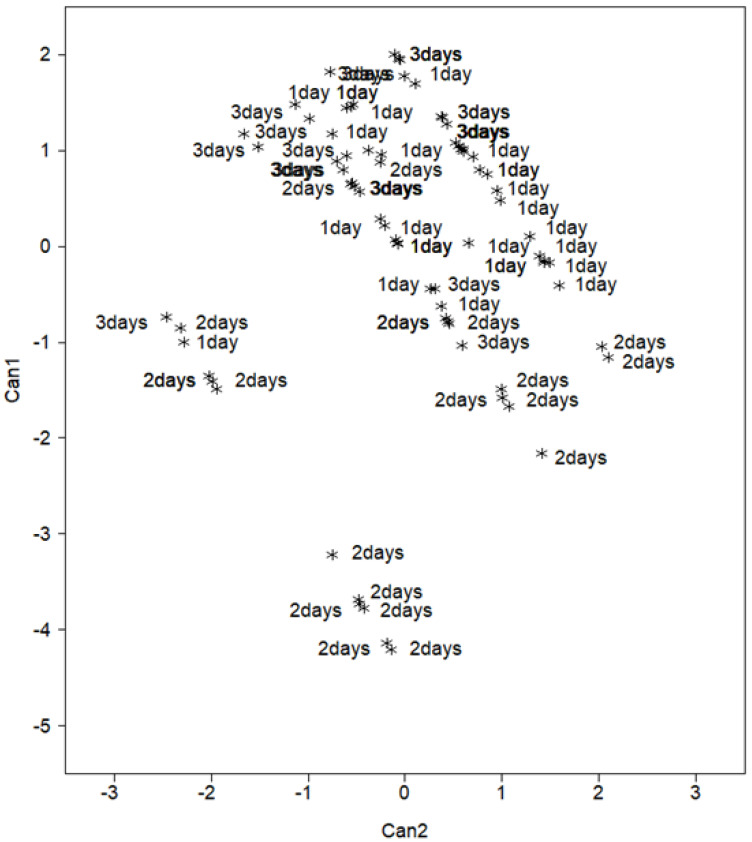
Scatter plot of centroids (class means) for the first and second canonical variates separating the fatty acid profiles of the northern Largemouth Bass whole body samples by diet (Trial 2). Treatments designated as 1: Fed daily (1 day), 2: fed every second day (2 days), and 3: fed every third day (3 days).

**Table 1 animals-12-02797-t001:** Growth metrics of northern Largemouth bass fingerlings (Trial 1).

Growth Metric	Fed 3%	Fed 5%	Fed 7%	*Pr* > F
Initial weight (g)	12.68 ± 0.21	12.72 ± 0.13	12.68 ± 0.04	0.98
Survival (%)	98.33 ± 1.67	98.33 ± 1.67	95.00 ± 2.89	0.49
Final biomass (g)	709.29 ± 19.26	843.17 ± 19.56	875.76 ± 74.82	0.09
Final avg. weight (g)	36.05 ± 1.49 ^a^	42.89 ± 1.11 ^ab^	45.94 ± 0.38 ^b^	0.013
Final avg. total length (mm)	138.36 ± 0.97 ^a^	144.46 ± 1.71 ^ab^	147.32 ± 2.93 ^b^	0.052
Weight gain (%)	240.74 ± 19.95 ^a^	237.21 ± 6.21 ^ab^	262.25 ± 20.12 ^b^	0.014
Feed Conversion Ratio (FCR)	0.83 ± 0.04 ^a^	1.21 ± 0.03 ^ab^	1.66 ± 0.20 ^b^	<0.01
Fulton’s Condition Factor (K)	1.36 ± 0.01	1.42 ± 0.03	1.43 ± 0.01	0.07

Values reported as Mean ± SE. Means within rows with different letters were significantly different.

**Table 2 animals-12-02797-t002:** Selected fatty acid composition of the diet and whole-body samples (% of total fatty acids by weight) from northern Largemouth bass fingerlings (Trial 1).

Fatty Acids		Diet	Initial Fish	Fed 3%	Fed 5%	Fed 7%	PSE	*Pr* > F
16:1	Palmitelaidic Acid	22.94	0.20	1.98 ^ab^	10.37 ^a^	0.17 ^b^	1.86	0.02
16:1, n-7	Palmitoleic Acid, n-7	8.20	10.10	9.31 ^ab^	8.90 ^b^	11.15 ^a^	0.28	0.04
17:1	Heptadecenoic Acid	0.93	1.23	0.28 ^a^	0.14 ^ab^	0.11 ^b^	0.02	0.03
18:2, n-6	Linoleic Acid, n-6	13.31	1.39	6.49 ^a^	3.90 ^ab^	1.24 ^b^	0.67	<0.01
18:3, n-3	Alpha-linolenic Acid	0.27	0.09	0.47 ^a^	0.25 ^ab^	0.13 ^b^	0.05	0.01
20:3	Eicosatrienoic Acid	0.18	0.09	0.27 ^a^	0.03 ^b^	0.00 ^b^	0.04	<0.01
22:1	Erucic Acid	0.64	1.20	0.35 ^a^	0.08 ^b^	0.04 ^b^	0.03	<0.01
20:5, n-3	Eicosapentaenoic Acid	7.17	0.38	1.14 ^a^	0.73 ^a^	0.16 ^b^	0.14	<0.01
22:5, n-3	Docosapentaenoic Acid	1.50	0.31	1.14 ^a^	0.66 ^ab^	0.06 ^b^	0.18	<0.01
22:6, n-3	Docosahexaenoic Acid	5.12	1.22	3.48 ^a^	2.31 ^a^	0.66 ^b^	0.40	<0.01
	Saturated ^1^	10.64	43.58	38.87 ^b^	37.59 ^b^	49.73 ^a^	1.72	0.01
	Monounsaturated ^2^	57.62	47.34	40.77	48.79	41.92	1.78	0.09
	PUFA ^3^	31.71	9.08	20.36 ^a^	13.62 ^b^	8.34 ^c^	1.42	<0.01
	LC-PUFA ^4^	15.50	2.38	6.39 ^a^	3.95 ^a^	1.19 ^b^	0.75	<0.01
	Total n-3 ^5^	14.06	2.00	6.23 ^a^	3.95 ^a^	1.02 ^b^	0.77	<0.01
	Total n-6 ^6^	15.55	3.56	9.20 ^a^	6.24 ^ab^	3.76 ^b^	0.67	<0.01
	n-3:n-6	0.90	0.56	0.68 ^a^	0.63 ^a^	0.27 ^b^	0.05	<0.01
	Total Lipid	16.26	25.59	29.79	23.96	26.60	1.02	0.46

All values are means of N = 3 replicate tanks of fish per weight group. Means within columns with different letters were significantly different (ANOVA, *p* ≤ 0.05). ^1^ Saturated fatty acids included 14:0, 15:0, 16:0, 17:0, 18:0, 19:0, 21:0, 22:0, 23:0, 24:0, and 25:0. ^2^ Monounsaturated fatty acids included 16:1, 17:1, 18:1, n-7, 20:1, 22:1 and 24:1. ^3^ Polyunsaturated fatty acids included 18:2, 18:2, n-6, 18:2 CLA, 18:3, n-6, 18:3, n-3, 20:2, 20:3, 20:4, 20:5, 22:2, 22:3, 22:4, 22:5, and 22:6. ^4^ LC-PUFAs (long chain polyunsaturated fatty acids) included 20:2, 20:3, 20:4, 20:5, 22:2, 22:3, 22:4, 22:5, and 22:6. ^5^ Total n-3 fatty acids included 22:5, 20:5, 18:3, n-3, and 22:6. ^6^ Total n-6 fatty acids included 22:2, 22:3, 20:2, 20:3, 20:4, 18:2 CLA, and 18:2 CL.

**Table 3 animals-12-02797-t003:** Class means on the first two canonical variates based on the fatty acid composition of the northern Largemouth bass whole-body samples of each class (Trial 1).

Class	Canonical Variates	
	CAN1	CAN2
Treatment 1: Fed 3%	1.491	−0.501
Treatment 2: Fed 5%	−0.198	1.273
Treatment 3: Fed 7%	−1.294	−0.77
Eigenvalue	1.39	0.87
Cumulative variance	0.61	1.00
Canonical correlation	0.72	0.07
*Pr* > F	<0.0001	0.005

**Table 4 animals-12-02797-t004:** Pooled within class standardized canonical coefficients (loadings) on the first two canonical variates estimated from the fatty acid profiles of northern Largemouth bass whole-body samples (Trial 1).

Fatty Acid	CAN1	CAN2
c16:1	0.027	0.906
c20:3	0.014	−1.051
c20:4	0.567	−0.191
c22:1	0.639	−0.377
c22:5	−1.450	−0.557
c22:6	1.936	1.887

**Table 5 animals-12-02797-t005:** Growth metrics of northern Largemouth bass fingerlings (Trial 2).

Growth Metric	Fed Daily	Fed Every 2nd Day	Fed Every 3rd Day	*Pr* > F
Initial weight (g)	7.15 ± 0.07	7.18 ± 0.03	7.14 ± 0.03	0.90
Survival (%)	97.78 ± 2.22	100.00 ± 0.00	97.78 ± 2.22	0.56
Final biomass (g)	356.78 ± 14.99 ^a^	272.26 ± 21.58 ^b^	183.14 ± 9.50 ^c^	<0.01
Final avg. weight (g)	24.39 ± 1.49 ^a^	18.15 ± 1.45 ^b^	12.47 ± 0.38 ^c^	<0.01
Final avg. total length (mm)	117.88 ± 1.80 ^a^	109.09 ± 2.19 ^b^	102.07 ± 0.85 ^b^	<0.01
Weight gain (%)	240.74 ± 19.95 ^a^	152.75 ± 34.29 ^b^	74.72 ± 5.40 ^c^	<0.01
Feed Conversion Ratio (FCR)	0.83 ± 0.03 ^a^	0.93 ± 0.03 ^a^	1.50 ± 0.15 ^b^	<0.01
Fulton’s Condition Factor (K)	1.49 ± 0.03 ^a^	1.39 ± 0.03 ^a^	1.17 ± 0.03 ^b^	<0.01

Values reported as Mean ± SE. Means within rows with different letters were significantly different.

**Table 6 animals-12-02797-t006:** Selected fatty acid composition of the diet and whole-body samples (% of total fatty acids by weight) from northern Largemouth bass fingerlings (Trial 2).

	Fatty Acids	Diet	Initial Fish	Fed Daily	Fed Every 2nd Day	Fed Every 3rd Day	PSE	*Pr* > F
16:1	Palmitelaidic Acid	22.94	0.20	17.10 ^b^	25.94 ^a^	14.43 ^b^	5.74	<0.01
16:1, n-7	Palmitoleic Acid	8.20	10.10	7.30 ^b^	10.78 ^a^	8.81 ^ab^	1.72	<0.01
18:3, n-3	Linolenic Acid	0.47	0.09	6.30 ^b^	5.74 ^ab^	8.72 ^b^	2.78	0.02
20:1	Eicosenoic Acid	2.06	2.80	0.71 ^b^	1.42 ^a^	0.76 ^ab^	0.61	0.04
20:5, n-3	Eicosapentaenoic Acid	7.17	0.38	1.32 ^ab^	0.76 ^b^	2.08 ^a^	1.22	<0.01
22:5, n-3	Docosapentaenoic Acid	1.50	0.31	0.82 ^ab^	0.61 ^b^	0.76 ^a^	1.02	<0.01
22:6, n-3	Docosahexaenoic Acid	5.12	1.22	4.54 ^b^	2.67 ^b^	2.87 ^a^	2.95	<0.01
	Saturated ^1^	10.64	43.58	10.77	6.21	8.54	2.89	0.20
	Monounsaturated ^2^	57.62	47.34	42.46	49.04	39.19	7.66	0.13
	PUFA ^3^	31.71	9.08	46.77	44.74	52.26	8.06	0.10
	LC-PUFA ^4^	15.50	2.38	10.21 ^b^	7.18 ^b^	13.56 ^a^	4.85	<0.01
	Total n-3 ^5^	14.06	2.00	14.93 ^b^	10.93 ^b^	21.00 ^a^	7.63	<0.01
	Total n-6 ^6^	15.55	3.56	2.42	2.22	1.75	1.19	0.30
	n-3:n-6	0.90	0.56	6.62	9.02	13.69	7.43	0.39
	Total Lipid	16.26	25.59	29.42	30.09	29.66	2.94	0.47

All values are means of N = 3 replicate tanks of fish per weight group. Means within columns with different letters were significantly different (ANOVA, *p* ≤ 0.05). ^1^ Saturated fatty acids included 14:0, 15:0, 16:0, 17:0, 18:0, 19:0, 21:0, 22:0, 23:0, 24:0, and 25:0. ^2^ Monounsaturated fatty acids included 16:1, 17:1, 18:1, n-7, 20:1, 22:1 and 24:1. ^3^ Polyunsaturated fatty acids included 18:2, 18:2, n-6, 18:2 CLA, 18:3, n-6, 18:3, n-3, 20:2, 20:3, 20:4, 20:5, 22:2, 22:3, 22:4, 22:5, and 22:6. ^4^ LC-PUFAs (long chain polyunsaturated fatty acids) included 20:2, 20:3, 20:4, 20:5, 22:2, 22:3, 22:4, 22:5, and 22:6. ^5^ Total n-3 fatty acids included 22:5, 20:5, 18:3, n-3, and 22:6. ^6^ Total n-6 fatty acids included 20:2, and 18:2 CL.

**Table 7 animals-12-02797-t007:** Class means on the first two canonical variates based on the fatty acid composition of the northern Largemouth bass whole body samples of each class (Trial 2).

Class	Canonical Variates	
	CAN1	CAN2
Treatment 1: Fed daily	0.40	0.29
Treatment 2: Fed every 2nd day	−1.79	−0.08
Treatment 3: Fed every 3rd day	0.99	−0.23
Eigenvalue	1.36	0.05
Cumulative variance	0.96	1.0
Canonical correlation	0.76	0.22
*Pr* > F	<0.0001	0.16

**Table 8 animals-12-02797-t008:** Pooled within class standardized canonical coefficients (loadings) on the first two canonical variates estimated from the fatty acid profiles of northern Largemouth bass whole body samples (Trial 2).

Fatty Acid	CAN1	CAN2
18:1 n9	0.26	0.19
18:1 n7	0.07	0.07
22:6	1.68	−1.10

## Data Availability

Data collected is available upon reasonable request from the authors of this study.

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
