# Peer review of "Effects of Different Feeding Regimes on Growth Rates and Fatty Acid Composition of Largemouth Bass Micropterus nigricans at High Water Temperatures"

_animals, 2022, doi:10.3390/ani12202797_

Round 1

Reviewer 1 Report

The subject of the current manuscript is surely worthy of investigation.  However, there are some points in the manuscript that I would like you to reconsider. I evaluate this manuscript as a major revision, as several points need to be carefully revised before the next resubmission as follows:

-          In the Abstract, the total fish number and fish number per replicate are missed. Also, add the average initial weight of fish ± SE. The duration of the experimental period should be clearly mentioned.

-          Old references have been cited, especially in the introduction section. Please update or remove.

-          Too many citations in the introduction (E.g., lines 43, 52, 62). Please minimize.

-          The hypothesis of the study should be clarified before the aim at the end of the Introduction section.

-          How was the water temperature controlled?

-          Line 136: The authors had to justify on what basis they selected these percentages.

-          The total fish number and fish number per replicate are missed in each trial.

-          How were the water quality parameters sampled?

-          Tables 1 and 5: It is suggested to calculate the final weight (g/fish) instead of the final biomass. How was weight gain (%) calculated? Also, add the data of feed intake. Move "Values reported as Mean ± SE. Means within columns with different letters were significantly different" to the Table footnote

-          In all tables: it should be “Means within a row …….. different". Also, describe the experimental groups and all abbreviations used in the tables' footnotes.

-          The number of references used in the discussion part is very few.

Author Response

In the Abstract, the total fish number and fish number per replicate are missed. Also, add the average initial weight of fish ± SE. The duration of the experimental period should be clearly mentioned.

Changes were made as requested.

Old references have been cited, especially in the introduction section. Please update or remove.

We believe the references to older research is integral to understanding how the LMB industry has progressed and evolved. The introduction is structured first to explain the industry challenges, then to focus more on nutrition, finally leading to the primary focus of this study. Since little research has been conducted, the use of older references is necessary.

Too many citations in the introduction (E.g., lines 43, 52, 62). Please minimize.

We believe all citations are relevant to the essence of this study.

The hypothesis of the study should be clarified before the aim at the end of the Introduction section.

The aim of the study is the hypothesis. We believe it would be unnecessary/redundant to use the term “hypothesis” or describe specific expected experimental outcomes as the reader may be bogged down.

How was the water temperature controlled?

Thank you, changes were made as requested. See updated manuscript (line 125)

Line 136: The authors had to justify on what basis they selected these percentages.

Changes were made as requested. See updated manuscript (line 172)

The total fish number and fish number per replicate are missed in each trial.

Tank numbers indicated in Methods sections 2.1 (line 149) and 2.5 (line 209), and Fish numbers indicated in Methods sections 2.2 (line 164) and 2.6 (line 221).

How were the water quality parameters sampled?

Changes made as requested.

Tables 1 and 5: It is suggested to calculate the final weight (g/fish) instead of the final biomass. How was weight gain (%) calculated? Also, add the data of feed intake. Move "Values reported as Mean ± SE. Means within columns with different letters were significantly different" to the Table footnote

Changes were made to include how weight gain % was calculated in section 2.7. The “Values reported as …” were moved to table footers as requested.

In all tables: it should be “Means within a row …….. different". Also, describe the experimental groups and all abbreviations used in the tables' footnotes.

Changes made as requested.

The number of references used in the discussion part is very few.

While having more references is desirable, the amount of research on largemouth bass feeding rates is minimal.  Therefore, we have added a few more references.

Reviewer 2 Report

These two experiments are of great significance to guide the feeding of largemouth bass during the high temperature season. The paper was well writing.  I have some suggetions: 

(1)If possible, provide  more detailed feed information, such as feed fomula, pellet size and properties.

(2)Provide basic nutrients of the test fish.

(3)Provide health indicators of the test fish.

(4)This experiment focuses on the influence of feeding strategy on fatty acids, so it is necessary to introduce the influence of feeding strategy on fatty acids in the introduction, and point out the significanece of this experiment  to detect fatty acids in test fish.

Author Response

(1) If possible, provide more detailed feed information, such as feed fomula, pellet size and properties.

We added the following in the methods- “3 mm” and “floating” pellet.  This is a commercial diet, and the company does not provide detailed ingredients.

(2) Provide basic nutrients of the test fish.

Not sure what the reviewer is asking. The company specifically formulated their feed based on the nutritional requirements of largemouth bass. We did not add any additional information concerning nutritional requirements.  If you are looking for proximate analysis, we did not do that as it was only a 28-day study. The initial fatty acid profiles for both trials are listed in tables 2 and 6

(3) Provide health indicators of the test fish.

Added: All fish were kept following IACUC protocols in methods section. Fish did not show signs of disease during the study in results section.

(4) This experiment focuses on the influence of feeding strategy on fatty acids, so it is necessary to introduce the influence of feeding strategy on fatty acids in the introduction, and point out the significance of this experiment to detect fatty acids in test fish.

Changes to the significance of fatty acids were made in the introduction as requested.

Reviewer 3 Report

Thank you for the opportunity to review manuscript J animals- 1920495. This research indicates that feeding LMB fingerlings at 3% of total body weight or feeding daily to satiation allows for efficient growth at 30℃and implements cost-effective feeding strategies., which is an important topic, and I commend the authors for taking on this research. The language of this manuscript is concise and logical. However, some issues in this manuscript must be improved. The paper will need major revision to be accepted, but I think you can do that. 1)Please provide full reasons for 30℃ as the study temperature. 2) There are some problems with the experimental method. 3)Adding more studies about fatty acids is suggested in the introduction to show the importance and necessity of studying. 4)Tables instead of large blocks of text are suggested to summarise experimental treatment so readers can quickly obtain experimental details. 5)The figures must be embellished to enhance their attractiveness and readability. Here are some detailed suggestions:

Simple Summary

Line 16: “feed amounts” is a part of “feed strategies”; consider rephrasing, please.

Line 17: “result in healthy fish growth” is better.

Abstract 

Line 23: There seems to be a mistake number in “Twenty LMB”.

Introduction

Adding more studies related to fish fatty acids and lipids is suggested in the introduction to show the importance and necessity of studying.

Paragraph 1: This study focuses on feeding strategies at high temperatures, so it is more important to introduce the ecological habits and proper growth temperatures of northern Largemouth bass than market requirements. Otherwise, it's hard to convince the reader that 30℃ would stress the fish. The marketing requirements section, while important, should be more concise.

Line 104: According to previous studies, 32℃does not seem to cause harmful effects on fish growth, and 30℃ is not the highest temperature the fish has ever experienced. Please provide the reason for choosing 30℃ as the experimental temperature in this study; otherwise, it looks like a random choice of temperature.

Materials and Methods

Line136-137: In my opinion, this feeding method has subjectively caused different daily feeding rhythms of fish in different experimental groups and even in different tanks, making the experimental data difficult to be convincing. Whether any previous studies have used this method or have shown that this method does not affect the ultimate growth of fish, please add specific details on determining the feeding amount for each feed. Please provide references for the process here. 

Line 185: Could you please add more experimental tips about “Any unconsumed feed was removed, counted, and added to the after-feeding weights to determine the total amount of feed consumed.”. How is “unconsumed feed” defined in this study? What is the weighing method?

Results

Line 236: “length” means “total length” or “body length”?

Line 236: According to table 1, there were significant differences only between the 3% and 7% feed rate treatments instead “the three treatments”.

Line 239-240: This sentence seems to be more appropriate in the context of experimental methods.

Table 1: It is necessary to provide the corresponding definition and calculation formula of the growth metric in Table 1.

Figure 1: Could you please add the confidence ellipse and axis contribution rate for figure 1?

Figure 2: Please add the confidence ellipse and axis contribution rate. Colours are recommended use to distinguish treatments.

Table 6: Add more statistics, such as F and P values.

Discussion

Line 339: From this study, the feeding strategy rather than the temperature affects its growth. It is not appropriate to say that 30℃affects growth without comparing the growth at normal temperature.

Line 344: How to determine that the amount of food being fed is "appropriate" and not prejudiced. I'm afraid I find it hard to agree with the feeding method mentioned in the Material and Methods section.

Conclusion

Line 395: This study shows that feeding strategy rather than temperature affects growth. Please consider rephrasing as no temperature gradient was set in this study.

Author Response

Simple Summary

Line 16: “feed amounts” is a part of “feed strategies”; consider rephrasing, please.

We believe rephrasing is not necessary. In Trial 1, the three experimental groups received three different feed amounts while maintaining an identical feeding protocol. In Trial 2, the three different experimental groups were subjected to three different feed strategies, which inherently resulted in the three respective experimental groups receiving different cumulative feed amounts by the conclusion of trial 2.

Line 17: “result in healthy fish growth” is better.

Changes were made as requested.

Abstract 

Line 23: There seems to be a mistake number in “Twenty LMB”.

Changes were made as requested. Fish numbers in abstract with respect to each trial were corrected.

Introduction

Adding more studies related to fish fatty acids and lipids is suggested in the introduction to show the importance and necessity of studying.

Changes were made by including more information and context on the importance and necessity of fatty acids and lipids in nutritional studies.

Paragraph 1: This study focuses on feeding strategies at high temperatures, so it is more important to introduce the ecological habits and proper growth temperatures of northern Largemouth bass than market requirements. Otherwise, it's hard to convince the reader that 30℃ would stress the fish. The marketing requirements section, while important, should be more concise.

We believe the background information provided in paragraphs 4 and 5 introduce temperature influences and address the reviewer's concerns about indicating that a temperature of 30 oC will stress LMB (reference #30). 

Line 104: According to previous studies, 32℃ does not seem to cause harmful effects on fish growth, and 30℃ is not the highest temperature the fish has ever experienced. Please provide the reason for choosing 30℃ as the experimental temperature in this study; otherwise, it looks like a random choice of temperature.

Please see response to previous comment

Materials and Methods

Line136-137: In my opinion, this feeding method has subjectively caused different daily feeding rhythms of fish in different experimental groups and even in different tanks, making the experimental data difficult to be convincing. Whether any previous studies have used this method or have shown that this method does not affect the ultimate growth of fish, please add specific details on determining the feeding amount for each feed. Please provide references for the process here. 

The specific feed amounts are provided in the method section.  The 3, 5, and 7% refers to body weight per day. We added sentences to the discussion that acknowledge the differences in feed rates to fish and the subsequent effect on growth. Since we used the industry percentages for Trial 1 as our feeding regime, the intent of the study was to determine at which percentage of body weight a day resulted in the best growth. We recognize that different feeding amounts would result in different growth rates.  But fish will only consume a certain amount of feed before the cost of net returns is more than economically viable. While producers could feed 5% or more to their fish, they are feeding excess amounts of feed that are not resulting in additional growth.

The satiation trial found feeding daily resulted in best growth rates.  We also found that the satiation feeding gave a 5.04% body weight per day as the best feeding option. This was confirmed in the 3, 5, and 7% study as it also provided the best growth.

Line 185: Could you please add more experimental tips about “Any unconsumed feed was removed, counted, and added to the after-feeding weights to determine the total amount of feed consumed.”. How is “unconsumed feed” defined in this study? What is the weighing method?

Changes made as requested.

Results

Line 236: “length” means “total length” or “body length”?

Thank you, changes made as requested. Corrected length to “total length” in all relevant sections.

Line 236: According to table 1, there were significant differences only between the 3% and 7% feed rate treatments instead “the three treatments”.

Changes made as requested

Line 239-240: This sentence seems to be more appropriate in the context of experimental methods.

Since this was a change made to the experimental protocol during trial 1, we believe it is more correct to include derivations to the methods in the results section, as these events occurred after the protocols were in place. Another example is the explanation of the nitrite concentration in section 3.3.  

Table 1: It is necessary to provide the corresponding definition and calculation formula of the growth metric in Table 1.

The formula for Fulton’s Condition Factor described in section 2.2, FCR and weight gain percentage formulas described in methods section 2.7

Figure 1: Could you please add the confidence ellipse and axis contribution rate for figure 1?

We believe the figure is able to illustrate the results of the discriminate analysis. Figures 1 and 2 are able to differentiate between the three treatments without including confidence ellipses at this current juncture.   

Figure 2: Please add the confidence ellipse and axis contribution rate. Colours are recommended use to distinguish treatments.

See response to the previous comment

Table 6: Add more statistics, such as F and P values.

In tables 1,2, 3, 5, 6, and 7, the column heading of "Pr>F" is the p-value associated with the F statistic of a given effect and test statistic.

Discussion

Line 339: From this study, the feeding strategy rather than the temperature affects its growth. It is not appropriate to say that 30℃ affects growth without comparing the growth at normal temperature.

Changes made as requested. 

Line 344: How to determine that the amount of food being fed is "appropriate" and not prejudiced. I'm afraid I find it hard to agree with the feeding method mentioned in the Material and Methods section.

Thank you, changes made as requested.

Conclusion

Line 395: This study shows that feeding strategy rather than temperature affects growth. Please consider rephrasing as no temperature gradient was set in this study.

Thank you, rephrasing changes made as requested

Round 2

Reviewer 1 Report

No further comments to be addressed

Reviewer 3 Report

Comments and Suggestions